# PKR Mediates the Mitochondrial Unfolded Protein Response through Double-Stranded RNA Accumulation under Mitochondrial Stress

**DOI:** 10.3390/ijms25147738

**Published:** 2024-07-15

**Authors:** Fedho Kusuma, Soyoung Park, Kim Anh Nguyen, Rosalie Elvira, Duckgue Lee, Jaeseok Han

**Affiliations:** 1Department of Integrated Biomedical Science, Soonchunyang University, Cheonan 31151, Republic of Korea; fedhokusuma@gmail.com (F.K.); ivydawn92@gmail.com (S.P.); nguyenkimanhdhd@gmail.com (K.A.N.); 2Soonchunyang Institute of Medi-Bio Science, Soonchunyang University, Cheonan 31151, Republic of Korea; maria.rosalie.elvira@gmail.com (R.E.); tomyoo27@gmail.com (D.L.)

**Keywords:** UPR^MT^, PKR, mitochondrial dsRNAs, integrated stress response, mitochondrial stress

## Abstract

Mitochondrial stress, resulting from dysfunction and proteostasis disturbances, triggers the mitochondrial unfolded protein response (UPR^MT^), which activates gene encoding chaperones and proteases to restore mitochondrial function. Although ATFS-1 mediates mitochondrial stress UPR^MT^ induction in *C*. *elegans*, the mechanisms relaying mitochondrial stress signals to the nucleus in mammals remain poorly defined. Here, we explored the role of protein kinase R (PKR), an eIF2α kinase activated by double-stranded RNAs (dsRNAs), in mitochondrial stress signaling. We found that UPR^MT^ does not occur in cells lacking PKR, indicating its crucial role in this process. Mechanistically, we observed that dsRNAs accumulate within mitochondria under stress conditions, along with unprocessed mitochondrial transcripts. Furthermore, we demonstrated that accumulated mitochondrial dsRNAs in mouse embryonic fibroblasts (MEFs) deficient in the Bax/Bak channels are not released into the cytosol and do not induce the UPR^MT^ upon mitochondrial stress, suggesting a potential role of the Bax/Bak channels in mediating the mitochondrial stress response. These discoveries enhance our understanding of how cells maintain mitochondrial integrity, respond to mitochondrial dysfunction, and communicate stress signals to the nucleus through retrograde signaling. This knowledge provides valuable insights into prospective therapeutic targets for diseases associated with mitochondrial stress.

## 1. Introduction

In the last few decades, mitochondria’s role in health and disease has gained interest exponentially. Aside from its well-known function as a cellular powerhouse, mitochondria possess other essential metabolic functions, such as calcium homeostasis and the synthesis of amino acids, fatty acids, and nucleotides [1,2,3,4]. Furthermore, mitochondria serve as signaling hubs to regulate programmed cell death and NLRP3 inflammasome activation [5,6]. To carry out these diverse set of functions, mitochondria contain over a thousand proteins, most of which are encoded by nuclear genes, imported from the cytosol, and properly folded in the mitochondria [7,8,9]. Thus, the disruption of its proteostasis could result in mitochondrial dysfunction. To preserve the mitochondrial homeostasis upon the stressful condition, cells activate a transcriptional response known as the mitochondrial unfolded protein response (mitochondrial UPR, UPR^MT^) [10].

UPR^MT^ is well established in *C. elegans*, where the activating transcription factor associated with stress (ATFS-1) is the key mediator for this response. ATFS-1 has a nuclear localization signal (NLS) as well as N-terminal mitochondrial-targeting sequences (MTS) [11]. In a normal condition, ATFS-1 is imported into the mitochondria and degraded by the protease LONP-1 [12]. However, under mitochondrial dysfunction conditions, the mitochondrial import of ATFS-1 is reduced, resulting in the accumulation of ATFS-1 in the cytoplasm and then it traffics to the nucleus driven by its NLS to activate the transcription of its target genes [13]. 

In contrast to the well-defined mechanism observed in *C. elegans*, the precise pathway by which the UPR^MT^ is activated in mammalian cells remains elusive. However, several recent studies identified ATF5 as a functional ortholog of ATFS-1 [14]. In addition to ATF5, ATF4 and CHOP are also suggested to be involved in the UPR^MT^ activation [15,16,17,18]. Beyond ATF5, ATF4, and CHOP, there has been considerable evolutionary divergence in UPR^MT^ signaling between *C. elegans* and mammalian cells. This divergence includes the expansion of transcription factors and the incorporation of additional signaling mechanisms. It is of note that the expression of all three transcription factors requires the phosphorylation of the alpha subunit of the translation initiation factor 2 (eIF2α) and subsequent signaling pathway, collectively referred to as the integrated stress response (ISR) [19].

The ISR is an evolutionary conserved signaling network for the cells to adapt to variable environmental and pathophysiological stresses, including a proteostasis defect, oxidative stress, nutrient deprivation, and viral infection [20]. These various stresses are sensed by four specific kinases responsible for eIF2α phosphorylation: PKR-like endoplasmic reticulum kinase (PERK), general control non-derepressible 2 (GCN2), protein kinase RNA-activated (PKR), and heme-regulated eIF2α kinase (HRI) [20]. The phosphorylated eIF2α inhibits the activity of eIF2B, a guanine nucleotide exchange factor for the eIF2-GDP complex, resulting in a global attenuation of protein synthesis [21,22]. Paradoxically, certain genes with upstream open reading frames (uORFs) in the 5′ UTR, such as ATF4 and CHOP, are preferentially translated [19]. These proteins function as transcription factors to induce various genes involved in regulating the stress response and fine-tuning the outcomes of these stresses [23].

PKR, a well-established antiviral sensor, has the primary function to sense and respond to double-stranded RNA (dsRNA), a molecular pattern often associated with viral infections [24]. Upon binding to dsRNA, PKR undergoes autophosphorylation and activation, leading to eIF2α phosphorylation and global protein synthesis, thereby limiting viral replication and activating stress response pathways [25]. Interestingly, PKR has been demonstrated to be activated in uninfected cells by cellular endogenous dsRNAs during mitosis [26], and further investigation revealed that mitochondrial RNAs constitute a major fraction of the PKR interactome [27]. On the other hand, mitochondrial dsRNAs have emerged as a potent damage-associated molecular pattern activating PKR and the ISR in several disease models [28,29,30]. While an earlier study has implicated PKR’s role in UPR^MT^ activation [31], the interplay of mitochondrial dsRNAs and PKR activation in the UPR^MT^ is yet to be addressed.

Recent studies have demonstrated increased ISR activation during mitochondrial stress [15], suggesting a potential role for eIF2α phosphorylation and its kinases. While several studies have proposed that GCN2 [32], PERK [33], HRI [34,35], and PKR [31] might sense mitochondrial dysfunction and transduce the signal to the ISR, the specific role of PKR in this process has not yet been elucidated. In this study, we revealed that PKR is involved in the activation of the UPR^MT^ by sensing mitochondrial dsRNAs generated under mitochondrial stress conditions.

## 2. Results

### 2.1. Critical Role of ISR in the Activation of UPR^MT^

Although several studies have suggested that the ISR plays a pivotal role in triggering the UPR^MT^ following mitochondrial stress, conclusive evidence directly linking the ISR to the UPR^MT^ remains elusive. Given the critical role of eIF2α phosphorylation in initiating the ISR, we investigated whether it is essential for UPR^MT^ activation under mitochondrial stress conditions. To this end, we utilized mouse embryonic fibroblasts (MEFs) that feature a homozygous alanine substitution at serine 51 of the eIF2α subunit (*Eif2a*^A/A^), effectively preventing its phosphorylation. 

To induce mitochondrial stress, we treated the MEFs with gamitrinib-triphenylphosphonium (GTPP), which inhibits the mitochondrial chaperone HSP90 homolog TRAP1 [36,37]. As a positive control to induce the ISR, we used thapsigargin (TG), a SERCA inhibitor known to induce ER stress and subsequent eIF2α phosphorylation. As expected, the *Eif2a*^A/A^ MEFs did not exhibit phosphorylated forms of eIF2α following treatment with GTPP and TG, in contrast to the increase in phosphorylated eIF2α observed in the wildtype *Eif2a*^S/S^ MEFs treated with both chemicals (Figure 1A). In the *Eif2a*^A/A^ MEFs, the expression of the UPR^MT^ genes such as *Hspd1* and *Hspe1*, as well as the ISR-related genes including *Ddit3*, *Gdf15*, and *Fgf21*, did not increase following treatment with either chemical, in contrast to the response observed in the *Eif2a*^S/S^ MEFs (Figure 1C).

Furthermore, we investigated the role of ATF4, which functions downstream of eIF2α phosphorylation in the ISR signaling pathway, in the mitochondrial stress response elicited by GTPP using ATF4 knock-out (*Atf4*^KO^) MEFs. In the absence of ATF4 protein expression in the *Atf4*^KO^ cells (Figure 1B), the expression of the UPR^MT^-related genes was not induced following treatment with either GTPP or TG (Figure 1D). This lack of response underscores the critical role of ATF4 in UPR^MT^ signaling.

The impact of the ISR on UPR^MT^ induction was further confirmed through the application of ISRIB, an inhibitor of the ISR signaling pathway. Consistent with previous observations, treatment with ISRIB significantly attenuated the expression of genes related to the UPR^MT^ and the ISR (Figure 1E). This corroborates the findings in *Eif2a*^A/A^ MEFs, where the absence of eIF2α phosphorylation hindered the necessary signal transduction to activate these genes, emphasizing the indispensable role of the ISR, and particularly ATF4, in the regulation of the UPR^MT^.

### 2.2. PKR Mediation of ISR and UPR^MT^ in Response to Mitochondrial Stress

The ISR is initiated by the phosphorylation of eIF2α, mediated by four kinases: GCN2, PERK, HRI, and PKR. While these kinases have been implicated in mitochondrial stress-mediated UPR^MT^ activation, the specific role of PKR in this process remains unclear. Given the close relationship between the ISR and UPR^MT^ induction following GTPP treatment, we focused on elucidating PKR’s contribution to ISR activation and the UPR^MT^ under mitochondrial stress conditions.

We employed MEFs derived from PKR knock-out (*Pkr*^KO^) mice to delineate the specific role of PKR in this pathway. To confirm the loss of PKR, we treated the MEFs with the viral dsRNA mimic poly(I:C) and assessed the expression of the *Ifnb1* gene, which is specifically activated by double-stranded RNA and is a known downstream target of PKR. In the *Pkr*^KO^ MEFs, phosphorylation of eIF2α was absent, and there was a failure to induce *Ifnb1* following transfection with the dsRNA analog poly(I:C), which is in marked contrast to the response observed in the PKR wildtype (*Pkr*^WT^) MEFs (Figure 2A). We then treated both the *Pkr*^KO^ and *Pkr*^WT^ MEFs with GTPP and TG. The absence of PKR in the *Pkr*^KO^ MEFs resulted in reduced eIF2α phosphorylation in response to GTPP, whereas there was no significant difference in the eIF2α phosphorylation between the *Pkr*^KO^ and *Pkr*^WT^ MEFs when treated with TG (Figure 2B). In addition, *Ifnb1* expression was induced in the *Pkr*^WT^ upon GTPP and TG treatment but not in the *Pkr*^KO^ MEFs (Figure 2C). This differential response suggests that GTPP-induced phosphorylation of eIF2α is contingent on the presence of PKR, highlighting its unique role in the cellular response to mitochondrial stress.

Furthermore, the expression of genes associated with the UPR^MT^ and ISR was not induced by GTPP in the *Pkr*^KO^ MEFs, as compared to the *Pkr*^WT^ MEFs (Figure 2D). This absence of gene induction in *Pkr*^KO^ MEFs confirms PKR’s critical involvement in mediating the mitochondrial stress response via the ISR pathway, thereby underscoring the kinase’s essential role in cellular stress signaling mechanisms.

### 2.3. Double-Stranded RNA Formation and Localization in Response to Mitochondrial Stress

After confirming that PKR is essential for transferring the signal to activate the ISR and subsequently induce the UPR^MT^, we sought to understand how mitochondrial stress could trigger PKR activation. PKR is traditionally known to be activated by dsRNAs originating from either exogenous or endogenous origin [26,38,39]. This led us to investigate potential sources of dsRNAs generated during mitochondrial stress.

To explore this, we first investigated the presence of dsRNAs in cells undergoing mitochondrial stress induced by GTPP treatment, employing the J2 antibody, which specifically recognizes dsRNAs within cells [40]. We observed the presence of dsRNAs recognized by the J2 antibody as early as one hour post-treatment (Figure 3A), with an increased puncta diameter of the J2 signal in a dose-dependent manner (Figure 3B). 

A further analysis on the localization of these dsRNAs showed that they overlapped with MitoTracker, a fluorescent dye that labels mitochondria in live cells (Figure 3A,B). An orthogonal view of a single cell treated with GTPP revealed overlapping signals between the J2 antibody (dsRNAs) and MitoTracker (mitochondria), indicating colocalization (Figure 3C). This finding suggests that the dsRNAs generated by GTPP treatment are located within the mitochondria.

To further explore the origin of dsRNAs upon GTPP treatment, we isolated mitochondrial RNAs from MEFs treated with either DMSO or GTPP for 2 h. These mitochondrial RNAs were then treated with strand-specific endoribonucleases to assess their structural characteristics. The treatment of mitochondrial RNAs from the GTPP-treated cells with RNase T1, which specifically targets single-stranded RNA, resulted in fewer degradation signals compared to those from the DMSO-treated controls (Figure 3D). Conversely, treatment with RNase III, which specifically targets dsRNAs, yielded more degradation signals in mitochondrial RNAs from the GTPP-treated cells compared to the controls (Figure 3D). This pattern indicates a higher abundance of dsRNA structures in the mitochondrial RNAs from cells exposed to GTPP, suggesting stress-induced alterations in RNA conformation.

### 2.4. Impaired Pre-Processing of Primary Transcripts and Mitochondrial dsRNA Accumulation

Mitochondrial RNA transcription, driven by mitochondrial RNA polymerase, produces primary transcripts composed of polycistronic units with multiple genes. These undergo detailed processing, including endonucleolytic cleavage, to separate individual tRNA, rRNA, and mRNA molecules. Disruptions in the assembly or function of mitochondrial ribonucleoprotein complexes responsible for RNA processing can lead to the accumulation of unprocessed or partially processed RNAs. Furthermore, due to bidirectional transcription from mitochondrial DNA’s light and heavy strands, such disturbances in mitochondrial proteostasis may lead to the generation of complementary RNA molecules, increasing the likelihood of dsRNA formation.

To validate this hypothesis, we initially assessed the impact of RNA transcription on the formation of dsRNAs during mitochondrial stress. Consistent with previous observations, pronounced dsRNA detection was evident through J2 antibody staining following treatment with GTPP (Figure 4A). The inhibition of transcription with Actinomycin D significantly reduced this staining, highlighting the dependence of dsRNA formation on active transcription under conditions of mitochondrial stress (Figure 4A). Importantly, ER stress induced by TG did not provoke J2 antibody staining, indicating that dsRNA formation is uniquely associated with disturbances in mitochondrial proteostasis (Figure 4A).

Further investigations concentrated on the status of pre-RNA processing, under the hypothesis that mitochondrial stress disrupts this essential function, leading to the accumulation of unprocessed RNA segments that may subsequently form dsRNA. To examine this, we conducted a quantitative analysis of the levels of unprocessed tRNA^Met^ using qRT-PCR. We specifically designed primers targeting the 5′ region of tRNA^Met^ adjacent to the tRNA^Leu^ region and the 3′ region adjacent to the *Nd1* region of the primary transcript (Figure 4B). These segments are typically processed by RNase P and RNase Z, respectively [41]. Following treatment with GTPP, there was a significant increase in the levels of these amplicons, indicating an increase in the amount of unprocessed tRNA^Met^ (Figure 4C). Conversely, treatment with TG did not result in a similar increase in unprocessed tRNA^Met^ levels (Figure 4C). This pattern of RNA processing, or the lack thereof, directly correlates with dsRNA formation, as detected by J2 antibody staining. This finding confirms the specificity of dsRNA formation to disruptions in mitochondrial proteostasis as opposed to ER proteostasis.

### 2.5. Cytosolic Translocation of Mitochondrial dsRNA and Its Role in PKR Activation

Next, we investigated whether dsRNAs accumulated in the mitochondria upon GTPP-induced stress could activate PKR and subsequently induce the UPR^MT^ in a manner similar to direct GTPP treatment. To this end, we isolated the total mitochondrial RNAs from the GTPP-treated and control DMSO-treated MEFs and transfected them into *Pkr^WT^* and *Pkr^KO^* MEFs to evaluate the induction of the UPR^MT^.

The mitochondrial RNAs isolated from the GTPP-treated MEFs significantly phosphorylated eIF2α, to a similar extent as observed with poly(I:C) treatment, in the *Pkr*^WT^ MEFs but not in the *Pkr*^KO^ MEFs, indicating a PKR-dependent mechanism (Figure 5A). Conversely, the mitochondrial RNAs from the DMSO-treated MEFs did not induce phosphorylation of eIF2α in either the *Pkr*^WT^ or *Pkr*^KO^ MEFs. Furthermore, the mitochondrial RNAs from the GTPP-treated MEFs triggered the expression of both UPR^MT^-related and ISR-related genes in the *Pkr*^WT^ MEFs but not in the *Pkr*^KO^ MEFs. Intriguingly, both the poly(I:C) and mitochondrial RNAs from the GTPP-treated MEFs induced the expression of *Ifnb1* to a much greater extent than the UPR-related genes exclusively in the *Pkr*^WT^ MEFs and not in the *Pkr*^KO^ MEFs (Figure 5B,C). These findings strongly suggest that the dsRNAs accumulated in mitochondria under GTPP-induced mitochondrial stress can activate the UPR^MT^ signaling pathway through PKR-dependent mechanisms.

Given the cytosolic localization of PKR and the mitochondrial accumulation of dsRNAs, we next explored the potential mechanism by which mitochondrial dsRNAs might activate PKR. We hypothesized that the accumulated mitochondrial dsRNAs could be released into the cytosol, thereby activating cytosolic PKR. To investigate this possibility, we examined the presence of mitochondria-originated dsRNAs in cytoplasmic RNAs isolated from cells treated with either DMSO or GTPP. Our findings revealed a significant increase in the cytosolic presence of mitochondrial transcripts, including *Nd1*, *Nd4*, *Nd5*, and *Cytb*, following GTPP treatment (Figure 5D). In contrast, TG treatment, which induces ER stress, did not result in an increase in these mitochondrial transcripts in the cytosol, suggesting that mitochondrial RNAs are specifically exported to the cytoplasm under conditions of mitochondrial stress induced by GTPP. This indicates a possible pathway for PKR activation through the release of mitochondrial dsRNAs into the cytosol during mitochondrial stress.

### 2.6. Bax/Bak Channels Export Mitochondrial dsRNAs upon Mitochondrial Stress

Next, we explored the mechanisms through which mitochondrial dsRNAs are translocated into the cytosol to activate PKR and the subsequent signaling pathways. Given that mitochondria are bounded by a double membrane, specific pathways must exist for the export of contents, including RNAs, into the cytosol. Previously, it was demonstrated that the Bax/Bak channels mediate the translocation of cytochrome c, which triggers apoptosis [42]. Subsequently, it was discovered that mitochondrial DNAs also utilize the Bax/Bak channels to reach the cytosol and induce an innate immune response by activating pathways, such as cGAS-STING [43,44]. More recently, several reports have suggested that mitochondrial RNAs can similarly exploit the Bax/Bak channels for their translocation into the cytosol [45,46,47].

Based on our understanding of mitochondrial dsRNA dynamics, we hypothesized that accumulated mitochondrial dsRNAs might be released into the cytosol via the Bax/Bak channel. To test this hypothesis, we employed Bax/Bak double-knock-out (*Bax/Bak*^dKO^) MEFs, treating them with GTPP to induce mitochondrial stress and subsequently monitoring the presence of mitochondrial RNAs in the cytosol. In the wildtype control (*Bax/Bak*^WT^) MEFs, a significant increase in cytosolic mitochondrial RNAs was observed (Figure 6A). However, this increase was absent in the cytosol of the *Bax/Bak*^dKO^ MEFs (Figure 6A), indicating a potential role of the Bax/Bak channels in mitochondrial RNA export. Furthermore, the phosphorylation of eIF2α, typically induced by GTPP treatment in the *Bax/Bak*^WT^ MEFs, was not observed in the *Bax/Bak*^dKO^ MEFs (Figure 6B). Similarly, the expression levels of the genes related to the UPR^MT^ and ISR were significantly reduced in the *Bax/Bak*^dKO^ MEFs compared to the *Bax/Bak*^WT^ MEFs (Figure 6C).

Importantly, there was no significant difference in the J2 staining, which indicates dsRNA presence, between the GTPP-treated *Bax/Bak*^WT^ and *Bax/Bak*^dKO^ MEFs (Figure 6D). This suggests that the diminished induction of the UPR^MT^ in *Bax/Bak*^dKO^ MEFs was not due to a reduced accumulation of mitochondrial dsRNAs. Moreover, when mitochondrial RNAs isolated from the GTPP-treated MEFs were used to induce the UPR^MT^ and ISR-related genes, no differences were observed between the *Bax/Bak*^WT^ and *Bax/Bak*^dKO^ MEFs (Figure 6E).

These results collectively suggest that the translocation of dsRNAs from stressed mitochondria through the Bax/Bak channels plays a crucial role in triggering PKR activation and the subsequent induction of the UPR^MT^. This underscores the importance of mitochondrial dsRNA export mechanisms in cellular stress signaling pathways.

## 3. Discussion

While the phosphorylation of eIF2α and the subsequent induction of ATF4 are established as critical mediators of mitochondrial stress signaling leading to the activation of the UPR^MT^ in mammalian systems, the specific pathways translating mitochondrial stress signals into cytosolic eIF2α phosphorylation remain unclear. In this study, we explored the role of PKR within this signaling cascade. Our findings indicate that in the absence of PKR, the UPR^MT^ is not initiated under conditions of mitochondrial stress, underscoring PKR’s essential role in detecting mitochondrial stress signals and mediating the phosphorylation of eIF2α. Furthermore, we observed that dsRNAs, which accumulate within mitochondria during stress, are transported into the cytosol where they activate PKR. Importantly, our data reveals that in the absence of the Bax/Bak channel, dsRNAs do not reach the cytosol nor trigger the UPR^MT^, identifying this channel as the crucial conduit for dsRNA translocation from the mitochondria to the cytosol. These findings collectively demonstrate that PKR is an indispensable kinase for phosphorylating eIF2α and initiating the UPR^MT^. This activation process is facilitated by the accumulation of dsRNAs within mitochondria and its subsequent release into the cytosol via the Bax/Bak channels under mitochondrial stress conditions (Figure 7).

Following the identification of ATFS-1 as a mediator of the UPR^MT^ in *C*. *elegans*, considerable research has been directed toward identifying a similar system to induce the UPR^MT^ in mammalian cells. It was found that the bZIP transcription factors ATF5, ATF4, and CHOP are associated with transcriptional activation of UPR^MT^-related genes [14,15,16,17,18]. Interestingly, these transcription factors are the downstream target of eIF2α phosphorylation, and studies have demonstrated that eIF2α phosphorylation and ISR induction are crucial for the initiation of the UPR^MT^ in response to mitochondrial stress. This raised questions about the mechanisms by which mitochondrial stress signals are relayed to the cytosol to trigger eIF2α phosphorylation. While it has been suggested that all four eIF2α kinases are implicated in UPR^MT^ activation upon mitochondrial stress, only HRI has been directly linked to activation by mitochondrial stress through its interaction with DELE1. Under mitochondrial stress, protease OMA1 is activated, and it cleaves DELE1 leading to its accumulation in the cytosol where it interacts with HRI, activating its eIF2α kinase activity [34,35]. Additionally, the crosstalk between mitochondria and the endoplasmic reticulum (ER) may activate PERK [48], while reactive oxygen species (ROS) generated during mitochondrial stress could potentially activate other kinases, including HRI, GCN2, and PKR [32,49,50]. Specifically regarding PKR, although several studies have implicated it in the mitochondrial stress-initiated UPR^MT^, the precise mechanisms remained unclear. Initially, PKR was discovered to mediate the UPR^MT^ induction in a DSS-induced colitis model [31], but exactly how PKR was activated was not explored. On the other hand, PKR was revealed to bind to endogenous dsRNAs, particularly of mitochondrial origin, and control eIF2α phosphorylation during mitosis and stress [26,27]. Subsequent studies demonstrated that mitochondrial dsRNAs in various disease models activate the ISR activation through PKR [28,29,30]. These findings suggest that mitochondrial dsRNAs might be the signal from stressed mitochondria to induce the UPR^MT^, although this correlation has not been definitively established. In this study, we provide evidence that PKR is crucial for activating the UPR^MT^ in response to mitochondrial stress induced by GTPP treatment. Furthermore, we observed that dsRNAs generated during GTPP treatment accumulate within the mitochondria, adding a new layer to our understanding of how mitochondrial distress signals are communicated to the cellular stress response machinery.

The presence of dsRNAs has traditionally been associated with viral infections, whether from RNA or DNA viruses. However, accumulating evidence suggests that dsRNAs can also be produced endogenously in various pathophysiological states [24]. In this study, we observed that GTPP-induced mitochondrial stress leads to the accumulation of dsRNAs within the mitochondria. Due to the circular form and bidirectional transcription of mitochondrial DNA, mitochondrial transcripts can form dsRNAs, which are typically recycled by a degradasome complex composed of the exoribonuclease polynucleotide phosphorylase (PNPase) and the ATP-dependent RNA helicase SUPV3L1 [51,52]. When the function of this degradasome is compromised, there is an increased formation of dsRNAs within the mitochondria [45,53]. Additionally, improper RNA processing of mitochondrial transcripts can also lead to the generation of mitochondrial dsRNAs [54]. GTPP, by inhibiting TRAP1, a mitochondrial chaperone, likely disturbs proteostasis, impairing the processing of mitochondrial transcripts and the degradation of unwanted dsRNAs by the degradasome [55]. In our study, GTPP treatment resulted in the accumulation of unprocessed mitochondrial tRNA^Met^, suggesting that disturbances in mitochondrial proteostasis may impair RNA processing mechanisms, thereby contributing to dsRNA generation under stress conditions. Furthermore, we found that halting transcription with actinomycin D eliminated dsRNA formation in mitochondria, indicating that newly synthesized primary transcripts are the main sources of dsRNAs.

In this study, we demonstrated that dsRNAs accumulate upon GTPP-induced mitochondrial stress and that these mitochondrial dsRNAs can activate PKR and subsequently induce the UPR^MT^ signaling pathway. Given that mitochondria are enclosed by a double membrane, the translocation of accumulated dsRNAs from the mitochondria to the cytosol requires specific gateways. Among the various channels in the mitochondrial outer membrane, the Bax/Bak channel protein is well known for facilitating the release of cytochrome c release to initiate apoptosis. In addition, recent publications suggest that mitochondria utilize this channel to translocate the macromolecules, including mitochondrial DNA [44]. Interestingly, some studies have demonstrated that mitochondrial dsRNAs are also released into the cytosol via the Bax/Bak channels [45]. Consistent with these findings, we observed that GTPP-induced mitochondrial stress could not induce PKR activation or subsequent UPR^MT^ signaling in the absence of Bax/Bak, even though dsRNAs clearly accumulated in *Bax/Bak*^dKO^ MEFs. In *Bax/Bak*^dKO^ MEFs, dsRNAs were not released into the cytosol. However, the accumulated dsRNAs in *Bax/Bak*^dKO^ MEFs were able to induce UPR^MT^ signaling when introduced into the cells by transfection, suggesting that the lack of UPR^MT^ induction in *Bax/Bak*^dKO^ MEFs is due to the blockade of dsRNA translocation. 

The findings in this study will enhance our understanding of how mitochondrial integrity is maintained and how stress signals are communicated to the nucleus via retrograde signaling. They provide a foundation for further research into therapeutic strategies targeting mitochondrial stress responses, which could be beneficial in treating diseases characterized by mitochondrial dysfunction. The identification of PKR and the Bax/Bak channels as key components in this pathway opens new avenues for developing interventions aimed at mitigating the effects of mitochondrial stress and enhancing cellular resilience. Furthermore, the role of dsRNA formation under mitochondrial stress conditions and its subsequent involvement in PKR activation and UPR^MT^ induction elucidates the intricate mechanisms of cellular stress responses and identifies potential molecular targets for therapeutic intervention.

## 4. Materials and Methods

### 4.1. Animal, Cell Culture, and Chemical Treatment

All the cells (listed in Table 1) were maintained in DMEM (Corning, NY, USA, 10-013-CV) supplemented with 10% (*v*/*v*) heat-inactivated fetal bovine serum (Corning, NY, USA, 30-002-CI) and 1% (*v*/*v*) penicillin/streptomycin (Corning, NY, USA, 35-015-CV) as described in a previous study [56]. The *Bax/Bak*^dKO^ MEFs cell line and its respective wildtype counterpart (*Bax/Bak*^WT^) were obtained from the American Type Culture Collections (ATCC, Manassas, VA, USA; CRL-2913, CRL-2907). The *Pkr*^KO^ mice were purchased from Australian Phenomics Facility (APF, APB631) at the Australian National University and were backcrossed to the C57BL/6 strain for at least six generations. The *Pkr*^WT^ and *Pkr*^KO^ mice were maintained in a 12 h dark/light cycle. All the animal care and experiment procedures were conducted according to the protocols and guidelines approved by Soonchunhyang University Animal Care and Use Committee (SCH23-0020). The primary *Pkr*^WT^ and *Pkr*^KO^ MEFs were isolated from the *Pkr*^WT^ and *Pkr*^KO^ mice, respectively, and immortalized by SV40 T antigen as previously described [23]. The MEFs were treated with DMSO, gamitrinib-triphenylphosphonium (GTPP; LegoChem Biosciences, Daejeon, Republic of Korea), thapsigargin (TG; Sigma, Livonia, MI, USA, T9033), ISRIB (Sigma, MI, USA, SML0843), and actinomycin D (Sigma, MI, USA, A9415) at the indicated concentrations and time points.

### 4.2. Subcellular RNAs Isolation and Transfection

Subcellular fractionation was performed using the Mitochondrial Isolation Kit for Cultured Cells (ThermoFisher, Carlsbad, CA, USA, 89874) according to the manufacturer’s instruction. The RNA was extracted from the cytoplasmic fraction and mitochondrial pellet using RiboEx™ LS (GeneAll, Seoul, Republic of Korea, 301-001) and RiboEx™ (GeneAll, Seoul, Republic of Korea, 302-001), respectively. The extracted RNA was then treated with RNase-free Turbo™ DNase (Invitrogen, Carlsbad, CA, USA, AM2238). For transfection of the mitochondrial RNAs, the MEFs were transfected in 12-well plates at 80% confluency with either low-molecular-weight poly(I:C) (Invivogen, CA, USA, TLRL-PICW) or mitochondrial RNAs at a concentration of 5 µg/mL using Lipofectamine™ RNAiMAX (Invitrogen, CA, USA, 13778075) with 1 µg:1 µL ratio. For the enzymatic treatment, 1 µg of nucleic acids was digested with 1 U of RNase T1 (Ambion, Naugatuck, CT, USA, AM2283) or RNase III (Ambion, CT, USA, AM2290) according to the manufacturer’s instructions. The isolated RNAs were run on 1.5% agarose gel and visualized by EtBr staining.

### 4.3. Reverse Transcription and qPCR Analysis

The total RNA was isolated from the indicated MEFs using RiboEx™ (GeneAll, Seoul, Republic of Korea, 302-001) and reverse transcribed using ReverTra Ace™ qPCR RT Master Mix (Toyobo, Tokyo, Japan, TOFSQ-201). A real-time qRT-PCR was performed using TOPreal™ SYBR Green qPCR PreMIX (Enzynomics, Daejeon, Republic of Korea, RT501M) on QuantStudio™ 1 (Applied Biosystems, Waltham, MA, USA). The primers used in this study are listed in Table 2.

### 4.4. Mitochondrial RNA Processing Analysis

The MEFs were seeded at 80% confluency on a 12-well plate and treated with GTPP (10 µM) or TG (300 nM) for 8 h. Total RNA isolation, cDNA synthesis, and qRT-PCR were performed as described above. Unprocessed mitochondrial tRNA^Met^ fragments were detected using primer pairs annealing at specific regions: (1) the 5′ end of tRNA^Met^ 5′ (forward) and an internal region of tRNA^Met^ (reverse), and (2) an internal region of tRNA^Met^ (forward) and 3′ end downstream of tRNA^Met^ (reverse). These primers specifically amplify the cleavage site of the RNase P at the 5′ end and RNase Z at 3′ end of tRNA^Met^, respectively. The Ct values from each target were normalized to the Ct values of total tRNA^Met^ obtained using internal primers for tRNA^Met^ (forward and reverse). The primers used for this analysis are listed in Table 2.

### 4.5. Immunoblotting

Protein lysates were prepared in RIPA lysis buffer (25 mM Tris-HCl pH 7.6, 150 mM NaCl, 1% NP-40, 1% sodium deoxycholate, and 0.1% SDS) supplemented with the Xpert protease inhibitor cocktail (Rockland, PA, USA, P3100) and Xpert phosphatase inhibitor cocktail (Rockland, USA, P3200). The protein concentrations were measured using the Pierce™ BCA Protein Assay Kit (ThermoFisher, CA, USA, 23225). A total of 10–20 µg of protein was mixed with Laemmli sample buffer (Bio-Rad, Hercules, CA, USA, BR1610747) containing 10% (*v*/*v*) 2-mercaptoethanol (Sigma Aldrich, Ann Arbor, MI, USA, M6250) and boiled at 100 °C for 5 min before loading onto an SDS-PAGE gel. The membranes were immunoblotted overnight at 4 °C with the following primary antibodies: ATF4 (Cell Signaling Technology, Danvers, MA, USA, #11815), p-eIF2α (Abcam, Fremont, CA, USA, ab32157), eIF2α (Cell Signaling Technology, MA, USA#9722), and Hsp90αβ (Santa Cruz, Dallas, TX, USA, sc-13119) using dilution according to the manufacturer’s recommendations. The antibody signals were visualized on medical X-ray films using the Agfa CP1000 automatic film processor. The band intensity was quantified by ImageJ software version 1.52.

### 4.6. Immunofluorescence Staining

The MEFs were seeded at 80% confluency on a 24-well cell imaging plate (Eppendorf, Framingham, MA, USA, 0030741021) one day before treatment. Prior to fixation, the MEFs were incubated with 5 µM MitoTracker™ Red CMXRos (ThermoFisher, CA, USA, M5710) for 15 min in the absence of serum. The MEFs were then washed twice with PBS and fixed with 4% (*w*/*v*) paraformaldehyde for 20 min at room temperature. Following fixation, the MEFs were washed three times with PBS and incubated with 0.3% Triton X-100 and 1% BSA in PBS for 45 min. The MEFs were then probed with J2 antibody (1:400, Scicons, 10010200) overnight at 4 °C. After washing three times with PBS, the MEFs were incubated with Alexa Fluor 488-conjugated secondary antibody (1:400; Jackson ImmunoResearch, West Grove, PA, USA, JAC-115-545-003) for 2 h at room temperature. Hoechst 33342 (ThermoFisher, CA, USA, H3570) was added during the final PBS wash to stain the nuclei. Confocal microscopy imaging was performed using Zeiss LSM 710. The yellow dsRNA puncta number and size were quantified using CellProfiler, software version 4.2.6.

### 4.7. Statistical Analysis

All the data values are presented as the means ± SD. The statistical significance of difference between the groups was assessed using the unpaired Student’s t test for single comparison and one-way or two-way ANOVA followed by Bonferroni’s post hoc test for multiple comparison. All the statistical analyses were performed using GraphPad Prism v9.5.0.

## Figures and Tables

**Figure 1 ijms-25-07738-f001:**
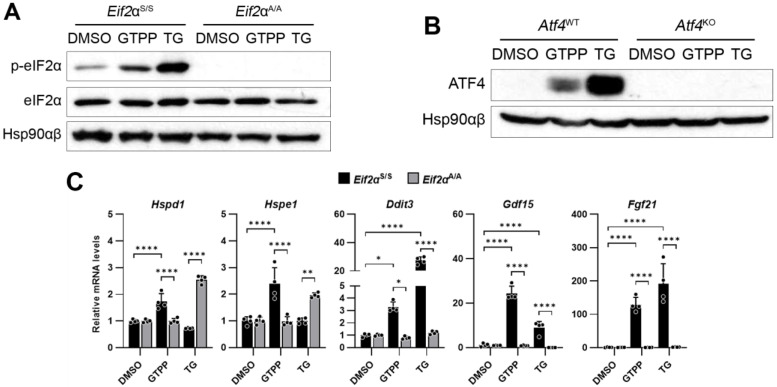
Integrated stress response (ISR) is required for mitochondrial unfolded protein response (UPR^MT^) activation. Indicated mouse embryonic fibroblasts (MEFs) were treated with mitochondrial chaperone TRAP1 inhibitor gamitrinib-triphenylphosphonium (GTPP; 10 μM), ER stressor thapsigargin (TG; 300 nM), or DMSO control. (**A**,**B**) Cell lysates were obtained 2 h (**A**) and 4 h (**B**) after stress induction for Western blot analysis. Hsp90αβ was used as loading control. (**C**–**E**) Total RNAs were isolated from (**C**,**D**) indicated MEFs or (**E**) wildtype MEFs treated in the presence of ISR inhibitor (ISRIB; 500 nM) at 8 h after stress induction for RT-qPCR analysis to measure UPR^MT^- and ISR-related genes (*n* = 4). Also, 18S rRNA primers were used as internal control. Data are presented as mean ± SD. * *p* < 0.0332, ** *p* < 0.0021, *** *p* < 0.0002, and **** *p* < 0.0001.

**Figure 2 ijms-25-07738-f002:**
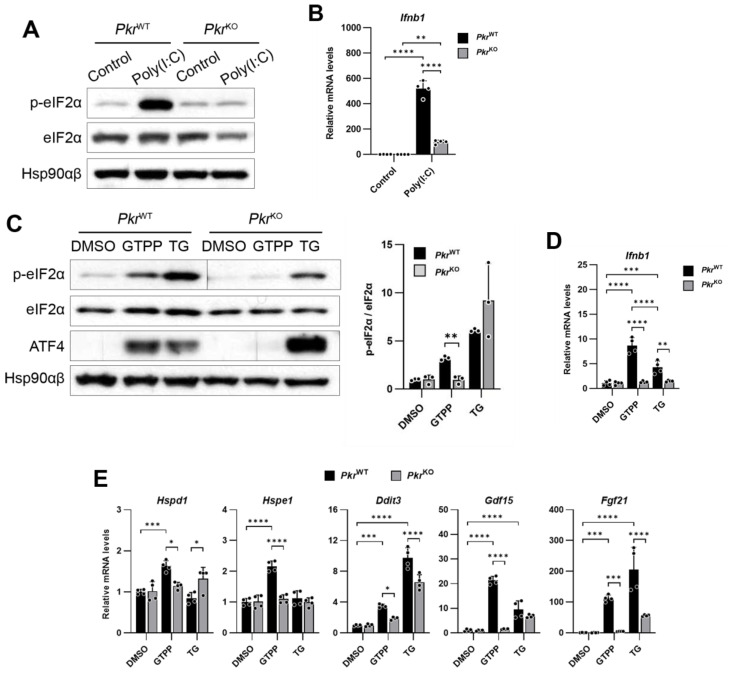
Immortalized *Pkr*^WT^ and *Pkr*^KO^ MEFs were generated from *Pkr*^WT^ and *Pkr*^KO^ mice. (**A**,**B**) Indicated MEFs transfected with double-stranded RNA (dsRNA) mimic poly(I:C) (5 µg/mL) were analyzed for Western blot (**A**) at 6 h post-transfection or RT-qPCR (**B**) at 12 h after transfection. (**C**,**D**) Indicated MEFs were treated with 10 μM GTPP and 300 nM TG up to 8 h. (**C**) Cell lysates were obtained 2 h after treatment for Western blot analysis. (**D**,**E**) Total RNA was isolated at 8 h after treatment for RT-qPCR analysis to measure *Ifnb1* and UPR^MT^- and ISR-related genes (*n* = 4). Hsp90αβ was used as loading control. Also, 18S rRNA primers were used as internal control. Data are presented as mean ± SD. * *p* < 0.0332, ** *p* < 0.0021, *** *p* < 0.0002, and **** *p* < 0.0001.

**Figure 3 ijms-25-07738-f003:**
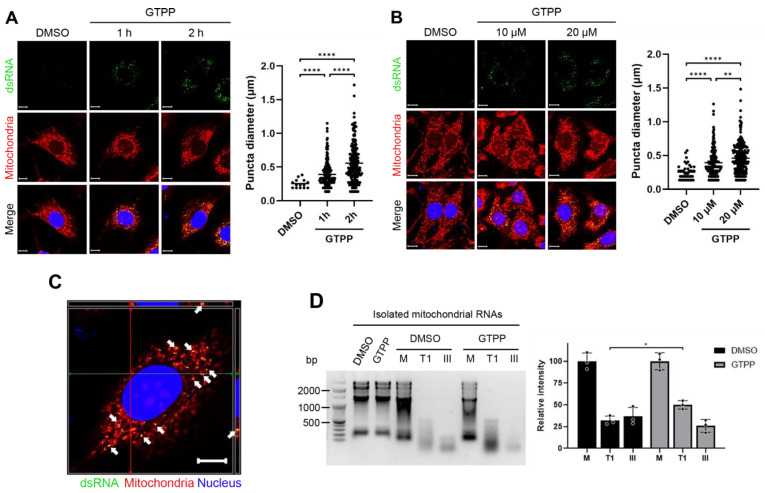
Wildtype MEFs were treated with mitochondrial chaperone TRAP1 inhibitor gamitrinib-triphenylphosphonium (GTPP) (**A**) at 10 µM concentration for indicated times or (**B**) at indicated concentrations for 1 h. Immunofluorescence staining was performed using monoclonal antibody J2 as marker for double-stranded RNAs (dsRNAs), MitoTracker Red CMXROS for mitochondria, and Hoecsht 33342 for nuclei. Yellow puncta diameter was measured across 3 images. (**C**) Orthogonal image of confocal microscopy on a single cell treated with GTPP was captured to observe complete localization, as represented by yellow puncta indicated by white arrows. Red and green lines indicate cross-section view on top and side panels. Scale bar, 10 µm. (**D**) Mitochondrial RNAs were isolated from wildtype MEFs treated with 20 µM GTPP or DMSO control for 1 h. The isolated RNAs were then digested using specific RNases. M, mock digestion. T1, single-stranded RNA-specific endonuclease RNase T1. III, dsRNA-specific endonuclease RNase III. Following digestion, samples were run on 1.5% agarose gel and visualized by EtBr staining (left). Densitometry was performed by measuring signal intensity normalized to the mock digestion (right). Data are presented as mean ± SD. * *p* < 0.0332, ** *p* < 0.0021, and **** *p* < 0.0001.

**Figure 4 ijms-25-07738-f004:**
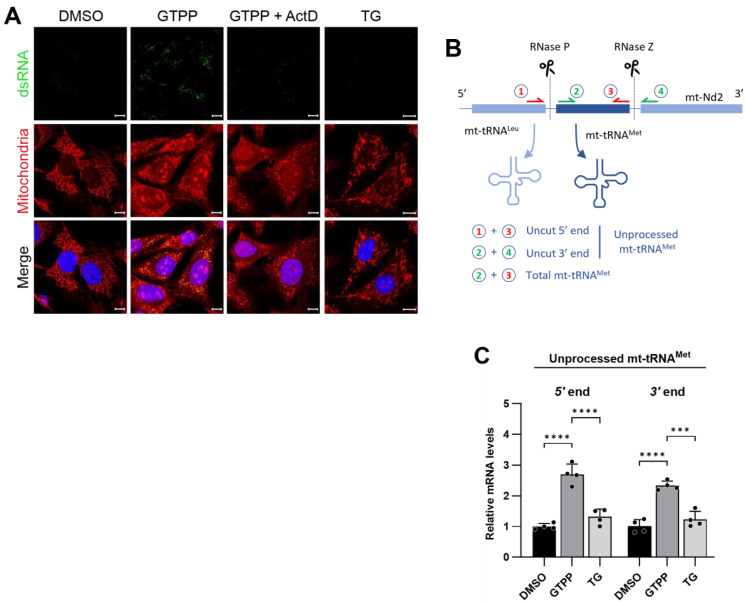
(**A**) Wildtype MEFs were treated with 10 µM GTPP alone or in the presence of transcription inhibitor actinomycin D (Act D; 15 µM) or 300 nM TG. Immunofluorescence staining was performed as described in Figure 3. (**B**) Schematic of mitochondrial polycistronic RNA processing represented by tRNA^Met^. RNase P and RNase Z cleave tRNA^Met^ at its 5′ and 3′ ends, respectively. (**C**) Total RNAs were isolated from wildtype MEFs at 8 h after treated with 10 µM GTPP, TG 300 nM, or DMSO control for RT-qPCR analysis to measure mitochondrial RNA processing. Unprocessed mitochondrial tRNA^Met^ fragments were measured by using primer pairs 1 and 3, or 2 and 4, as shown in (**B**) [for details, please refer to Materials and Methods Section 4.4]. The resulting Ct values were normalized to the Ct values of total mitochondrial tRNA^Met^ as obtained from primer pair 2 and 3 (*n* = 4). Data values presented as mean ± SD. *** *p* < 0.0002, and **** *p* < 0.0001.

**Figure 5 ijms-25-07738-f005:**
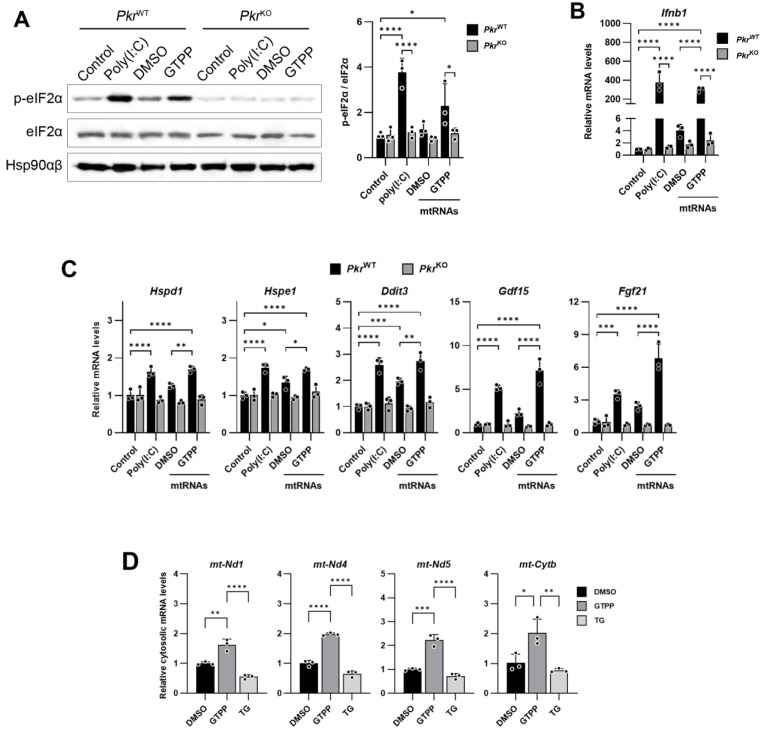
Mitochondrial RNAs were isolated from wildtype MEFs treated with 20 µM GTPP or DMSO control for 1 h and were then used to transfect into indicated MEFs. Poly(I:C) was used as positive control. (**A**) Cell lysates were obtained 6 h after transfection for Western blot analysis. Hsp90αβ was used as loading control. (**B**,**C**) Total RNAs were isolated 12 h after transfection for RT-qPCR analysis to measure (**B**) dsRNA-responsive *Ifnb1* and (**C**) UPR^MT^- and ISR-related genes. (*n* = 4). (**D**) Wildtype MEFs were treated with 20 µM GTPP, 300 nM TG, or DMSO control. Total RNAs were isolated from cytosolic fraction at 1 h after stress induction for RT-qPCR analysis to measure mitochondrial transcript levels (*n* = 3). Data values presented as mean ± SD. * *p* < 0.0332, ** *p* < 0.0021, *** *p* < 0.0002, and **** *p* < 0.0001.

**Figure 6 ijms-25-07738-f006:**
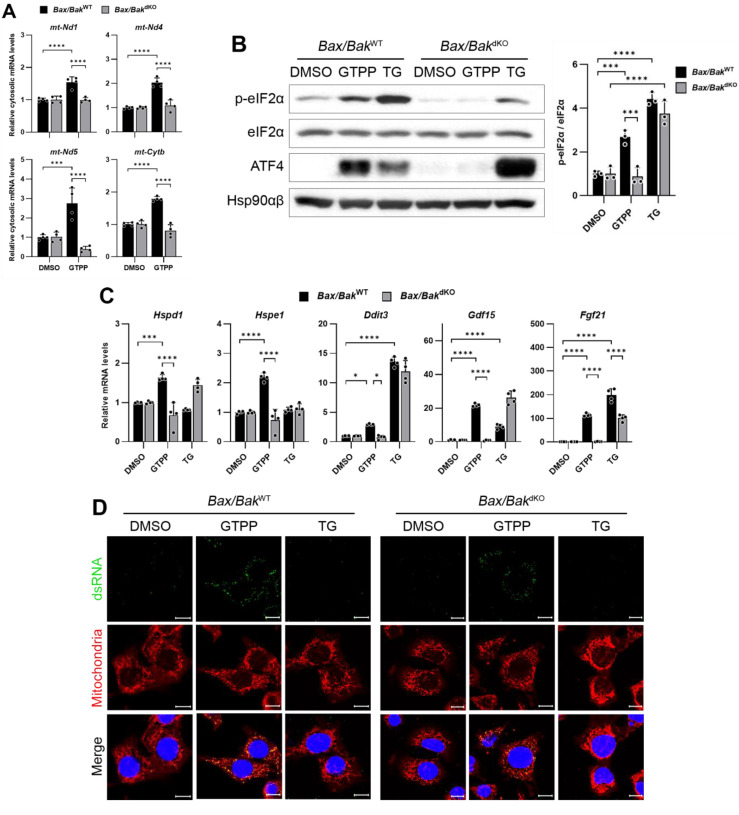
(**A**) Indicated MEFs were treated with mitochondrial chaperone TRAP1 inhibitor gamitrinib-triphenylphosphonium (GTPP; 10 µM) or DMSO control. Cytosolic RNAs were isolated at 8 h after stress induction for RT-qPCR analysis to measure mitochondrial transcript levels. (**B**,**C**) Indicated MEFs were treated with 10 µM GTPP, thapsigargin (TG; 300 nM), or DMSO control. (**B**) Cell lysates were obtained 2 h after stress induction for Western blot analysis. Densitometry was performed by measuring p-eIF2α signal normalized to total eIF2α. Hsp90αβ was used as loading control. (**C**) Total RNAs were isolated at 8 h after stress induction for RT-qPCR to measure UPR^MT^- and ISR-related genes (*n* = 4). (**D**) Immunofluorescence staining was performed 1 h after stress induction to observe dsRNA accumulation. (**E**) MEFs were transfected with mitochondrial RNAs isolated from GTPP- or DMSO-treated cells, and total RNAs were isolated 12 h after transfection for RT-qPCR analysis to measure UPR^MT^- and ISR-related genes (*n* = 3). Data values presented as mean ± SD. * *p* < 0.0332, ** *p* < 0.0021, *** *p* < 0.0002, and **** *p* < 0.0001.

**Figure 7 ijms-25-07738-f007:**
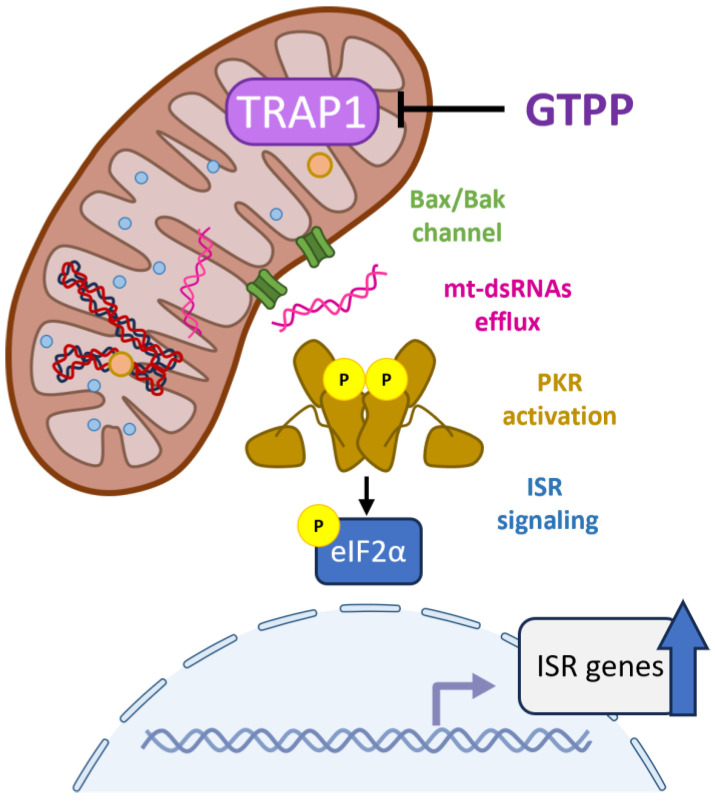
Role of mitochondrial dsRNAs and PKR in mitochondrial stress signal relay for UPR^MT^**.** When mitochondrial proteostasis is disturbed, mitochondrial RNA processing is impaired, leading to the accumulation of double-stranded mitochondrial RNAs. These dsRNAs are then exported to the cytosol via the Bax/Bak channel. In the cytosol, PKR binds to these mitochondrial dsRNAs and activates the integrated stress response (ISR) signaling cascade by phosphorylating eIF2α leading to the upregulation of ISR target genes.

**Table 1 ijms-25-07738-t001:** MEFs cell lines used in this study.

Cell Line	Modification	Reference(s)
*Eif2a* ^S/S^	Wildtype MEFs	[57]
*Eif2a* ^A/A^	Serine residue substitution at position 51 to non-phosphorylatable alanine	[57]
*Atf4* ^WT^	Wildtype MEFs	[58]
*Atf4* ^KO^	Targeted deletion in exon 1–3 of *Atf4* locus	[58]
*Pkr* ^WT^	Wildtype MEFs	this study
*Pkr* ^KO^	Targeted deletion in exon 2–3 of *Pkr* locus	this study, [59]
*Bax/Bak* ^WT^	Wildtype MEFs	[60,61,62]
*Bax/Bak* ^dKO^	Targeted deletion in exon 2–5 of *Bax* locus and exon 3–6 of *Bak* locus	[60,61,62]

**Table 2 ijms-25-07738-t002:** Primers used in this study.

Primers	Sequence
*18S rRNA* forward	CGC TTC CTT ACC TGG TTG AT
*18S rRNA* reverse	GAG CGA CCA AAG GAA CCA TA
*Hspd1* forward	AGT GTT CAG TCC ATT GTC CC
*Hspd1* reverse	TGA CTG CCA CAA CCT GAA G
*Hspe1* forward	GCG AAG GCG AGA GTC ATG
*Hspe1* reverse	TGC TTG CAA CAC TTT TCC TTG
*Ddit3* forward	CTG CCT TTC ACC TTG GAG AC
*Ddit3* reverse	CGT TTC CTG GGG ATG AGA TA
*Gdf15* forward	AGT GTC CCC ACC TGT ATC G
*Gdf15* reverse	TGT CCT GTG CAT AAG AAC CA
*Fgf21* forward	AGA TCA GGG AGG ATG GAA CA
*Fgf21* reverse	TCA AAG TGA GGC GAT CCA TA
*Nd1* forward	TCC GAG CAT CTT ATC CAC GC
*Nd1* reverse	GTA TGG TGG TAC TCC CGC TG
*Nd4* forward	TAA TCG CAC ATG GCC TCA CA
*Nd4* reverse	CAT TTG AAG TCC TCG GGC CA
*Nd5* forward	CAG CAC AAT TTG GCC TCC AC
*Nd5* reverse	TAG TCG TGA GGG GGT GGA AT
*Cytb* forward	TGC ATA CGC CAT TCT ACG CT
*Cytb* reverse	AGG CTT CGT TGC TTT GAG GT
*mt-tRNA^Met^* 5′ end upstream forward	GGA ATT GAA CCT ACA CTT AA
*mt-tRNA^Met^* internal 5′ end forward	AGT AAG GTC AGC TAA TTA AG
*mt-tRNA^Met^* internal 3′ end reverse	TAG TAC GGG AAG GAT TTA AAC
*mt-tRNA^Met^* 3′end downstream reverse	AGG ACC TAA GAA GAT TGT GA

## Data Availability

Data available on request from the authors.

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
