# Peer review of "PKR Mediates the Mitochondrial Unfolded Protein Response through Double-Stranded RNA Accumulation under Mitochondrial Stress"

_ijms, 2024, doi:10.3390/ijms25147738_

Round 1
Reviewer 1 Report
Comments and Suggestions for Authors
Using primarily WB and RT-PCR, the authors examined the molecular mechanism of the mitochondrial stress response with an emphasis on PKR and dsRNA. If the authors' stated number of data points is accurate, then reproducibility has been sufficiently confirmed. However, I believe that questions need to be raised regarding the suitability of the WB loading control and the precision of the quantification method, and I believe that enough more evidence is required before the paper can be accepted. The following issues must, at the very least, be suitably addressed with the help of earlier studies or new information.
Fig. 1 and further WB data: Although HSP90αβ is displayed as a loading control, isn't this improper given that HSP90 expression levels differ based on the type of stress? Is it known that the stress treatment used in this study has no effect on the expression levels of eIF2α or HSP90? Please provide evidence to support your explanation.
Figure 2: The legend is really unclear. Is there a way you could restructure it so it makes more sense? Is sample B taken two hours or eight hours following drug treatment? Or does the duration of treatment vary based on the drugs? If the duration of treatment and the time at which samples are collected vary based on the drug or assessment technique, please think about clearly illustrating this with a flowchart or something similar. Alternatively, please include a brief notice of the goal and these temporal discrepancies in the main text. Does Ifnb1 differ from UPRMT- and ISR-related genes in terms of lead time for induction?
Figure 2C: Have you verified that Pkr KO has no effect on the expression level of eIF2α, as shown in Figure 2C?
Line 161: "with an increase in detection of dsRNAs in a dose-dependent manner (Figure 3B)"
I am unable to feel the difference the authors discribed in the Confocal images. Could you please quantify something? The puncta counts within responding cells appear to include not so much defference. Whether the percentage of responsive cells relative to total cells changes cannot be assessed from the presented field alone.
Line 167: "This finding suggests that the dsRNAs generated by GTPP treatment are localized within the mitochondria."
Is there enough proof that the dsRNAs are entering the mitochondria rather than only existing on their surface? Is the Ifnb1 induction brought on by the dsRNAs not a nuclear or cytoplasmic event?
Figure 3C: Why do the reconstructions from the side and top views have differing numbers of yellow puncta? If you are displaying a cross-section of a particular area, please label the area you are displaying in the top view with a line.
Figure 4A: There is insufficient loading control and the HSP90 bands are fused together. The eIF2α bands do not appear to have same intensity in the data presented. Since the stress-responsive HSP90 is not very significant, please think about adjusting the amount of application such that the eIF2α bands have nearly same intensitiy if you would like to compare the ratio of eIF2α to p-eIF2α.
Figure 6B: The band intensity of eIF2α is very different. Dividing relative intensity ratios with bands like this can be very inaccurate. In the data presented, it appears that the intensity of p-eIF2α is lower in the dKO TG compared to the WT TG. However, the quantitative data (p-eIF2α / eIF2α) shows a trend towards an increase in the TG dKO side, which calls into question the accuracy of the quantification method.
Figure 7: It seems to contain excessive speculation. The direct interaction and activation of PKR with mt-dsRNAs has not been investigated. I believe that it has been shown that mt-dsRNAs promote signaling downstream of PKR. Since the effects of ATF4 on the same gene are different in GTPP and TG treatment, it is possible that there is more complex regulation downstream of ATF4. This figure may contain misleading information.
minor points
Every figure: The letters pointing to each panel are difficult to see, so I would like them to be improved. For example, please indicate them in the top left corner of each panel using a larger font.
line 145: (GTPP; 10 μM). for 8 hours.
-> (GTPP; 10 μM) for 8 hours.
Figure 3D: Please indicate which bands correspond to which base pairs next to the marker lane.
Figure 6D: Please think about doing a more suitable processing on the image. The fact that the GTPP-treated area is the only one with a signal is quite difficult to observe.
Reviewer 2 Report
Comments and Suggestions for Authors
The authors found that mitochondrial UPR (UPRmt) does not occur in the cells lacking Protein Kinase R (PKR) and suggest that PKR has a crucial role in phosphorylating eIF2α and initiating UPRmt. They observed a mitochondrial accumulation of double-stranded RNA during stress. Double-stranded RNA were not released in the cells deficient in the Bax/Bak channels, and UPRmt was not induced upon stress.
The authors conclude that Bax/Bak channels are a “critical gateway” for the release of dsRNAs into the cytosol during mitochondrial stress (“that dsRNAs are exported to the cytosol via the Bax/Bak channel”).
1. However, direct proof of Bax/Bak involvement in the process is missing. The authors should provide more support for their claim of the direct role of the two proteins in double-strand RNA translocation.
2. Although the essential literature on PKR and mitochondrial dsRNA is cited (in Discussion), the Introduction is based on the older literature. The existing knowledge on PKR role and that of mitochondrial double-stranded RNA in stress response should be described in the introduction.
Minor:
3. Figure 3D legend (add abbreviations used in the figure)
4. A table with explanations of MEFs modifications (Methods section)
5. Line 354, please omit “shown to be”
6. Lines 370-71, typo “was revealed binds”
Comments on the Quality of English LanguageMinor.
Round 2
Reviewer 1 Report
Comments and Suggestions for Authors
The authors addressed my comments very carefully, and the quality of their wording and data presentation have greatly improved. I believe that the present version is suitable for publication in IJMS.